# Do We Need Palliative Care in Pediatric Nephrology? Patients’ and Caregivers’ Disease Experience

**DOI:** 10.3390/children10020324

**Published:** 2023-02-08

**Authors:** Nina Kubiak, Chiara Fehrenbach, Jenny Prüfe, Julia Thumfart

**Affiliations:** 1Department of Pediatric Respiratory Medicine, Immunology and Critical Care Medicine and Cystic Fibrosis Center, Charité Universitätsmedizin Berlin, 13353 Berlin, Germany; 2Department of Pediatric Gastroenterology, Nephrology and Metabolic Diseases, Charité Universitätsmedizin Berlin, 13353 Berlin, Germany; 3Paediatrics II and III, University Hospital Essen, 45147 Essen, Germany

**Keywords:** palliative care, chronic kidney disease, children

## Abstract

Chronic life-limiting illnesses such as chronic kidney disease (CKD) require integral support to the families concerned in addition to medical care. Palliative care is an option to facilitate families to address future concerns, such as procedures for acute life-threatening complications, or to relieve physical and psychosocial suffering. The exact needs of patients or parents have not yet been investigated. To assess needs in supportive palliative care, we conducted a monocentric qualitative interview study. We included patients 14 to 24 years old as well as parents of younger children (below 14 years) with CKD ≥ stage 3. In total, fifteen interviews were conducted. Data were analyzed with a deductive and descriptive approach using qualitative content analysis as described by Mayring. Sociodemographic data and basic information of disease were collected using questionnaires. In contrast to caregivers, adolescents and young adults do not express worries about their own mortality or reduced life expectancy. Rather, they report about their limitations to everyday life associated with the disease, especially in the areas of school and work. They wish to live a normal life. Caregivers are concerned about the future and the disease trajectory. They also describe difficulties in balancing the management of the disease with other responsibilities such as work and healthy siblings’ needs. Patients and caregivers appear to need a chance to talk about their everyday struggles and disease-related fears and concerns. Talking about their concerns and needs may help deal with their emotions and facilitate acceptance of their situation characterized by a life-limiting disease. Our study confirms the need for psychosocial support in pediatric nephrology to address the needs of the affected families. This can be offered by pediatric palliative care teams.

## 1. Introduction

Many children, adolescents and young adults with chronic life-limiting conditions such as chronic kidney disease (CKD) experience significant limitations to their everyday life due to medical procedures such as dialysis, medication intake and frequent but often unscheduled hospital admittances [1]. Their health-related quality of life evaluated by self-report or by their caregivers is significantly lower as compared to children with other chronic diseases [2,3]. Sociodemographic and psychosocial factors (e.g., ineffective coping strategies) are important determinants of an impaired health-related quality of life [2]. Yet, routine clinical care of patients and their families often neglects the above-mentioned issues. 

Implementation of palliative care in these families may be a way to close this gap. Pediatric palliative care is understood as active and comprehensive care that takes into consideration the mind and soul of the child. It starts with the diagnosis and is continued beyond the death of the child by providing support to the family [4]. Palliative care should be provided to all children who suffer from life-limiting diseases and should be offered already when the bad news are broken [5,6]. 

The simultaneous offer of palliative care and life-sustaining treatments such as dialysis should be considered within a comprehensive strategy for both critically and non-critically ill patients [7,8].

Pediatric palliative care can relieve physical and psychosocial suffering, help with decision making and assist with the coordination of the multidisciplinary care team. The need for supportive palliative care may vary throughout the course of the disease with intensified support at the moment when the diagnosis is conveyed, in crises and in the terminal stage [9,10]. For patients, palliative care may help cope with concerns about school and friends. Pediatric palliative care includes also direct support for the family, for instance, by providing respite care for parents, enabling them to spend time with their other children.

In summary, palliative care goes beyond end-of-life care, which is what many understand by palliative care. 

Children, adolescents and young adults with CKD face lifelong increases in morbidity and mortality [11]. In pediatric nephrology, routine integration of palliative care is lacking [9] but should be implemented following practitioners’ opinions and experience [12]. However, we are unaware of any research in the field directly asking for CKD patients’ or their caregivers’ expectations and demands. Therefore, we conducted a qualitative study aiming to identify the point of view of adolescent and young adult patients as well as caregivers of younger children with advanced CKD.

## 2. Materials and Methods

### 2.1. Data Collection

The study took place in the department of Pediatric Gastroenterology, Nephrology and Metabolic Diseases, Charité Universitätsmedizin Berlin, Germany. The research protocol was approved by the institutional ethics committee. Participants were selected using convenience sampling procedures. Following the study design, we included patients aged between 14 and 24 years as well as parents of younger children (below 14 years) with CKD ≥ stage 3. All participants provided written informed consent. Fifteen interviews were conducted between May and November 2021. 

### 2.2. Methods

A semi-structured interview guideline was developed, and its applicability was assessed through pilot interviews (see Appendix A and Appendix A) [13]. Interviewees’ sociodemographic data and basic information on their disease were collected using questionnaires (Table 1). The interviews were conducted by N.K. and C.F. The interviews lasted for 15−50 min and were audio-recorded and transcribed verbatim according to the rules of Dresing and Pehl. MAXQDA© software was used to code the material and form categories and subcategories. Data were analyzed with a deductive and descriptive approach using qualitative content analysis, as described by Mayring [14]. One author performed the data analysis, and the other authors participated in and monitored the coding process. The results were examined and discussed by all authors. Quotations were translated by J.T. All quotations were back-translated into German by J.P. and compared with the original quotations by N.K. and C.F. to ensure that the intended meaning was retained.

## 3. Results

According to the qualitative research approach, the results are presented in a descriptive-interpretative form and illustrated with citations. In this way, the perspective of the interviewees is presented in a comprehensive way. First, there is an overview of the main categories and subcategories formed.

### 3.1. Main Categories

Based on a scoping literature review and our professional experience, we developed an interview guideline that reflects the main five categories. We then analyzed how these categories were filled with the patients’ and caregivers’ thoughts and feelings, thus deriving the subcategories from the transcripts (Table 2). A selection of categories with quotes is presented to highlight essential aspects from the relevant perspectives and to compare them with the other perspectives (parent vs. patient).

### 3.2. Caregivers’ View—Medical History (Breaking the Bad News)

Informing the patient and/or parents about the diagnosis of a chronic progressive kidney condition seems to be a highly stressful situation for many parents. Some parents express that conveying the diagnosis triggered strong emotional reactions up to “fear of death”. 


*“I just cried. I couldn’t take it at all that they told me, yes she has nephroblastoma [kidney tumor]. As if it wasn’t enough that she had a tumor, they said that not only one kidney, but the other one didn’t work neither, because the tumor was already on the other artery. And later it turned out that it was a de novo mutation, not inherited from the family. But to see this, this disease in your own child, yes, it’s hard.” *

*(Female caregiver, 31 years, child transplanted)*



*“But, I have to say that the conversation was difficult. She told us the diagnosis, we asked how bad is it, and she immediately said: “Well, he is probably a transplant patient, a dialysis patient, you have to assume.” If you have never dealt with this before, of course, it is for you, as a parent, you think, you are scared to death.” *

*(Female caregiver, 35 years, child on dialysis)*


### 3.3. Patients’ View—Medical History (Family Support in Coping with Disease)

Some young adults report that they feel supported by their parents in dealing with the disease. Often, parents divide the workload among each other. Commonly, one parent mainly takes on most tasks in managing the disease such as dialysis, outpatient visits and medication intake, while the other parent goes to work. This distribution can lead to the adolescents and young adults feeling a special bond and closeness to the “disease manager” parent.


*“Well, I grew up with four brothers and my parents, so to speak, six of them, and my dad was always the disease manager. He created the most blatant Excel tables with medications and always knew when we had to re-order medication. He did the dialysis until I was able to do it myself, so to speak. He also went on a school trip with me, […] and always came in the evening and took care of everything. He was a very close reference person for me. My mother was also always there, but took over less of the hospital things.” *

*(Woman, 22 years, transplanted)*



*“So my mother studied biology and she knows the most about it and was able to fight for me the most when it came to things that happened in the hospital. And she usually went with me to the doctor, because her job makes her more flexible than my father, for example. He can’t just say, by the way, I’m not here for a few hours.” *

*(Woman, 22 years, CKD stage 4)*


### 3.4. Caregivers’ View—Quality of Life

Caregivers describe difficulties in balancing disease management with their other responsibilities such as work, family and healthy siblings’ needs. Particularly in the case of hospitalization, the chronically ill child absorbs most of parental attention. Caregivers are faced with siblings’ feeling of being neglected.


*“Our son had to take a back seat in many areas. When we were in hospital a lot, for example, and everything revolved around X (his sister). And of course that was awful for Y (our son), because he was also going through puberty. And then there are sometimes these sentences in retrospect: “You weren’t there.” But then we were always compensating either way. Either, we were in hospital, couldn’t work and had to compensate balance things at home. And that was quite difficult sometimes.” *

*(Male caregiver, 53 years, child transplanted)*


Some parents report how important social networks are for them. These networks are regarded as a valuable part of practical coping. For example, caregivers may experience support from friends when they need help with balancing all responsibilities. 


*“But we are also lucky that, if necessary/ that we had our friends’ backs, that we were always able to solve it somehow.” *

*(Male caregiver, 53 years, child transplanted)*


### 3.5. Patients’ View—Quality of Life 

Adolescents and young adults worry less about their own mortality and more about the limitations associated with the disease in everyday life, especially in the areas of school and work. Examples from the interviews are listed below:


*“Well, what I am thinking is: there are always diseases worse than mine. So I am lucky. (…) I’m going to school and I’m doing eleventh grade and I would like to study pharmacy. (...) But, ‘cause of dialysis, I do not have that much time, so, hope I can do it. (...) Last school year, I only slept for about four or three hours per night because I missed so much and had to catch up on the material.” *

*(Woman, 18 years, on dialysis)*



*“And then I made it to the job interview and then they somehow still didn’t want me. Once, because I still did not have a driver’s license and second because of my kidney disease. (...) Yes, I thought that was a bit mean somehow.“ *

*(Man, 21 years, CKD stage 4)*


Specific procedures in CKD (e.g., dialysis access or stomas) can also affect how patients perceive their own body image. Patients are both ashamed and open about externally visible changes to their bodies. 


*“(...) that it was difficult and uncomfortable for me to talk about it [catheter] when someone asked. During sports lessons and when changing or so (...) Then I told it to the people individually, but I didn’t really tell it in a big way, only to the closer people, so to speak.” *

*(Boy, 16 years, transplanted)*



*“Beauty is, I think I am the most beautiful girl in the world, but, [patient and interviewer laugh] yes, my hand, where I was punctured, has really many scars and I don’t like that so much, yes. Still, I am not ashamed of it at all. Yes, and if someone asks me what it is, then I simply tell them what it is [fistula].” *

*(Woman, 18 years, on dialyses)*


Young people claim that living with their disease rather strengthened them. Dealing with the disease supported their personal development and their personality. This is also reflected in their social relationships. 


*“I think, I lost my–at the time–best friend also a little bit because of the disease. I think you also go through a bit of a personality change. However, I am actually quite happy about it, because, I think, it has changed a lot for the better in terms of personal development.” *

*(Woman, 22 years, CKD stage 4)*



*“Yes, I have changed since dialysis. I actually became stronger and my personality, too, a lot has changed.” *

*(Woman, 18 years, on dialyses)*


Overall, it seems to be significant for patients to live a normal life.


*“So far, I do not really have any problems with the disease. (...) I hardly notice the disease at all. Nothing hurts me or anything like this. I only notice that I am ill, because I have to go to the doctor all the time and because I have to take the medicines. It is just the knowledge that maybe in 10 or 20 years I may have to go on dialysis when my kidneys won’t work at all anymore. Otherwise, well, I actually live a very normal life.” *

*(Man, 19 years, CKD stage 3)*


### 3.6. Caregivers’ View—Thinking about the Future (Life Expectancy)

Compared to the patients, parents appear more concerned about the future and the future course of their child’s disease. They worry about life expectancy and quality of life. Hope is given, for example, by the prospect of future research.


*“[I worry about] dialysis or organ transplantation, whether it will all work out as one is hoping, so that it will be good for him [son] afterwards. And of course, this whole prognosis for him now in terms of life expectancy and so on [I’m also worried about that]. So, I just hope, I strongly hope for the research, that it will somehow progress further and further.” *

*(Female caregiver, 37 years, child CKD stage 4)*



*“We used to have a relatively easy life. We did not have certain worries. Now we do occasionally have very big worries, especially when we do not know what is going to happen next. [...] And when you realize that your daughter’s life expectancy, for example, is considerably reduced and her quality of life is also limited, then you really worry.” *

*(Male caregiver, 53 years, child transplanted)*


### 3.7. Patients’ View—Thinking about the Future (Starting a Family)

Female adult patients report that they are ambivalent or uncertain about family planning. They are worried about how a pregnancy can affect the further course of their disease. Family planning is a difficult topic for some young adult women; some may avoid thinking about it.


*“Otherwise, I would like to have children of my own, but I am also afraid that my kidney will deteriorate as a result. That the values will deteriorate. And I don’t think I could cope with that, because you never know whether another transplant will be possible, whether there will be an offer, whether it will be just as good.” *

*(Woman, 22 years, transplanted)*



*“About family. I don’t know. I think patients try to suppress it a bit.” *

*(Woman, 22 years, CKD stage 4)*


### 3.8. Caregivers’ View—Satisfaction with Care System (Medical Care)

Caregivers expressed feeling challenged by changing doctors and the discontinuity of care, particularly when hospitalized.


*“I sometimes find it difficult when you are on the ward and always have a different contact person. Sometimes it’s an assistant doctor, sometimes it’s a senior doctor, sometimes it’s Dr. X” *

*(Female caregiver, 35 years, CKD Stage 5).*


### 3.9. Patients’ View—Satisfaction with Care System (Psychological Support)

Some patients indicated a wish to talk to psychologists about their feelings associated with the disease. Communication with third parties about physical characteristics and strategies for dealing with bullying should also be addressed.


*“Probably it would have been good then too/ from today’s point of view I think it would have been good if you had somehow received psychological care (...) that you could somehow talk about how you feel (...) and come to terms with it, and also discuss it when people make comments.” *

*(Woman, 22 years, CKD stage 4)*


## 4. Discussion

This is the first study examining the perspective of patients and caregivers regarding aspects of palliative care in pediatric nephrology. The aim of the study was to gain insights into how patients and parents cope with the challenges of a life-limiting illness and, if necessary, how they can be supported in managing these challenges. 

According to our local ethics committee, no direct questions on topics such as advanced care planning or end of life were asked in the interview. However, in answering open questions about the future, caregivers explicitly addressed themes such as morbidity and life expectancy and revealed struggling with emotions. Dealing with these worries is part of the palliative care approach. Parents start worrying about the child’s survival as soon as the diagnosis is delivered. Therefore, palliative support for the family should also begin at this point, as postulated in the IMPaCCT statement [4]. 

It is the first study that allowed adolescents with CKD to speak for themselves on this topic. Patients often depend on the support of their parents in their disease management. In most cases, one parent takes over the role of a “disease manager” for the sick child. From the patient’s point of view, a special bond develops with this parent.

Parents report the difficulty of balancing all tasks and duties (at work, in the family). Other studies addressing the quality of life of parents of children on dialysis demonstrated impaired well-being and a high burden on the caregivers [15,16]. 

It can happen that a healthy sibling might feel neglected in everyday life because the parents’ attention is absorbed by the care for the sick child. In our study, this could only be explored from the parents’ perspective as we did not interview siblings. In the study by Tong et al., parents of children with CKD also reported disrupting family norms such as sibling neglect [17]. The impaired quality of life of the unaffected siblings of children with life-limiting diseases is reported [18] and especially among siblings of kidney transplant recipients [19]. In the context of palliative care, counseling could include the entire family with the healthy siblings to give information and provide coping strategies [20]. This intervention is likely to improve the functioning of the entire family. Of course, not every family is dealing with questions of life expectancy or suffering. In many cases, integration of a psychosocial child health specialist will be sufficient. However, the transitions between psychosocial support and palliative care support should be fluent. 

There are parents who can fall back on the support of friends in difficult situations. Agerskov et al. ascertained the importance of friends for parents of children with CKD. Friends can promote practical help in everyday life and emotional support [21].

In our study, patients are concerned about daily hassles at school or at work. Lifestyle limitations of young adults with CKD are also reported by Kerklaan et al. [22]. Cosmetic changes (such as a catheter) make some young people feel insecure. Other young people deal with the disease in a self-confident way and find themselves attractive despite visible changes to their bodies. Some patients feel stronger through experiences in coping with CKD [23]. Friendships can change due to the course of the disease. Most patients shared their wishes to live a normal life and to be seen as a person rather than a chronically ill patient as reported by Coombes et al. [12].

In our study, patients’ and caregivers’ views differ on some topics. Caregivers explicitly address themes such as morbidity and life expectancy. Patients, however, do not actively address their limited life expectancy, and it remains unclear whether or not they are aware of their health condition being a life-limiting one.

Rather, patients are more concerned about struggles in their daily life at school or at work. However, we did not interview patients and parents from the same family. 

Young adults are worried about their future in terms of family planning. Communication with others about the disease should also be addressed in professional conversations.

Talking more about these issues could probably help caregivers and patients to deal with their emotions and worries and could facilitate acceptance of a situation characterized by a life-limiting disease of their children. In many pediatric nephrology centers, social workers and psychologists offer psychosocial support to affected families [24]. From our point of view, talking about such topics goes beyond primary psychosocial care. A fixed contact person on the primary care team would be useful for patients and caregivers. This person should be educated in palliative care such as a special palliative care curriculum.

Limitations to our study are single-center recruitment and selection bias as only patients and caregivers with higher educational levels were interested in participating in the study, thus our sample not being representative. Although the total cohort of 15 participants is adequate for a qualitative study, it should be mentioned that the cohort of patients is rather small with *n* = 5. Moreover, we only interviewed adolescents aged 14 years and above. The lower age limit was suggested by our local ethics committee because this was the first study to directly interview children and adolescents with chronic kidney diseases.

## 5. Conclusions

Our findings here indicate the need for palliative care structures in pediatric nephrology to address the needs of the affected families [6]. These structures could help to open communication on topics of life limitation. The same holds probably true for all pediatric disciplines that deal with children with life-limiting conditions. Further research is needed to investigate patient needs and health professional resources.

## Figures and Tables

**Table 1 children-10-00324-t001:** Demographic characteristics of the interview participants.

		Caregiver (*n* = 10)	Patient (*n* = 5)
Sex	Female/male	6/4	3/2
Age	Median age [range] in years	50,5 [29–54]	19 [16–22]
Age of patient at the time of diagnosis	At time of birth <6 months ≥6 months	3 3 4	2 3 0
Stage of CKD	Stage 3–4 Stage ESKD Dialysis Transplantation	4 1 5	2 1 2
Marital status	Single Married Divorced	0 9 1	5 0 0
Citizenship	German Not German	6 4	4 1
Religion	Christian Muslim None	2 3 5	0 1 4
Place of residence	Berlin City Rural	6 4	3 2
Degree of disability of the patient	<50% ≥50% None	0 9 1	1 4 0
Care level of the patient	None Care level 1–2 * Care level 3–4 **	4 3 3	4 1 0
Graduation of interview partner	Secondary school Middle school High school	1 3 6	0 0 5
Professional status of interview partner	Housewife/-man Education Academic education Unemployed Attending school	2 3 4 1 0	0 1 2 0 2
Monthly net income of interview partner	<EUR 1.000 EUR 1.000–3000 >EUR 3.000 No income	4 2 4 0	2 0 0 3

* Minor impairments of independence or abilities (grade 1) and considerable impairments (grade 2). ** Severe impairments (grade 3) and most severe impairments (grade 4).

**Table 2 children-10-00324-t002:** Overview of main categories and subcategories.

Main Categories	Subcategories
Medical History	Request for further medical information about disease Breaking bad news Transmitting diagnosis Family support in coping with disease
Coping with disease	Resources Barriers
Quality of life	Siblings Social relationship, peer group Research Free time Cosmetic changes in the body School/Education/Academic career Discipline dealing with disease Vacation/Holidays
Thinking about the future	Life as couple/starting a family Life expectancy
Satisfaction with care system	Language barriers Medical care Transition Psychological support Psychosocial care/advice

## Data Availability

The data presented in this study are available upon reasonable request from the corresponding author. The data are not publicly available because participants of this study did not agree for their data to be shared publicly.

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
