# Peer review of "Do We Need Palliative Care in Pediatric Nephrology? Patients’ and Caregivers’ Disease Experience"

_children, 2023, doi:10.3390/children10020324_

Round 1

Reviewer 1 Report

The authors undertook a single-center qualitative study involving interviews with a small cohort of CKD patients ages 14-22 years and a separate group of parents caring for younger (< 14 years of age) CKD patients regarding medical history, coping with disease, quality of life, thinking about the future, and satisfaction with the care system. Comments are provided in the attached pdf. The authors are commended for exploring a very important set of issues in a particularly vulnerable population.

The background and discussion need to be expanded significantly. In particular the relationship between the presented results and pediatric palliative care (the focus as currently written) seems tenuous at best. The very small sample size is a significant limitation that should be clearly mentioned. The rationale for highlighting various themes (particularly when the sample size is so small) need to be mentioned - particularly if there were any a-priori decision points. Finally, comparing and contrasting the current results with similar studies in the adult CKD patient population would help highlight the unique needs of the pediatric population. 

Reviewer 2 Report

It is very important theme for health care workers ,but very difficult to reveal clearly.

1: How about patient's financial situation?  Financial situation influence patient's character.

2:Why did you select the patients between 14 and 24 years old? Is it difficult to express own thinking below 14 years old patients?

Reviewer 3 Report

 A good job about palliative care in pediatric nephrology. The main limitations of this study are the limited number of patiets (n=5) and single center recruitment, which may lead to selection bias in patients and caregivers. Authors should collect sufficient cases to support their conclusion. The main results were language based instead of data.

Round 2

Reviewer 3 Report

None.

Author Response

Thanks for your comment.